# Gut-Liver Axis, Gut Microbiota, and Its Modulation in the Management of Liver Diseases: A Review of the Literature

**DOI:** 10.3390/ijms20020395

**Published:** 2019-01-17

**Authors:** Ivana Milosevic, Ankica Vujovic, Aleksandra Barac, Marina Djelic, Milos Korac, Aleksandra Radovanovic Spurnic, Ivana Gmizic, Olja Stevanovic, Vladimir Djordjevic, Nebojsa Lekic, Edda Russo, Amedeo Amedei

**Affiliations:** 1Faculty of Medicine, University of Belgrade, 11000 Belgrade, Serbia; ankica.vujovic88@gmail.com (A.V.); aleksandrabarac85@gmail.com (A.B.); milos.korac@med.bg.ac.rs (M.K.); spurnic@yahoo.com (A.R.S.); stevanovicolja74@gmail.com (O.S.); vladimir.djordjevic@kcs.ac.rs (V.D.); nesalekic67@gmail.com (N.L.); 2Hospital for Infectious and Tropical Diseases, Clinical Center of Serbia, 11000 Belgrade, Serbia; gmizic_ivana@yahoo.com; 3Faculty of Medicine, Universisty of Belgrade; Institute of Medical Physiology “Rihard Burijan”, 11000 Belgrade, Serbia; mdjelic011@gmail.com; 4Clinic for Digestive Surgery, Clinical Center of Serbia, 11000 Belgrade, Serbia; 5Department of Experimental and Clinical Medicine, University of Florence, 50134 Florence, Italy; edda.russo@unifi.it (E.R.); amedeo.amedei@unifi.it (A.A.); 6Department of Biomedicine, Azienda Ospedaliera Universitaria Careggi (AOUC), 50134 Florence, Italy

**Keywords:** gut microbiota, gut-liver axis, chronic liver diseases, fecal transplantation, probiotics

## Abstract

The rapid scientific interest in gut microbiota (GM) has coincided with a global increase in the prevalence of infectious and non-infectivous liver diseases. GM, which is also called “the new virtual metabolic organ”, makes axis with a number of extraintestinal organs, such as kidneys, brain, cardiovascular, and the bone system. The gut-liver axis has attracted greater attention in recent years. GM communication is bi-directional and involves endocrine and immunological mechanisms. In this way, gut-dysbiosis and composition of “ancient” microbiota could be linked to pathogenesis of numerous chronic liver diseases such as chronic hepatitis B (CHB), chronic hepatitis C (CHC), alcoholic liver disease (ALD), non-alcoholic fatty liver disease (NAFLD), non-alcoholic steatohepatitis (NASH), development of liver cirrhosis, and hepatocellular carcinoma (HCC). In this paper, we discuss the current evidence supporting a GM role in the management of different chronic liver diseases and potential new therapeutic GM targets, like fecal transplantation, antibiotics, probiotics, prebiotics, and symbiotics. We conclude that population-level shifts in GM could play a regulatory role in the gut-liver axis and, consequently, etiopathogenesis of chronic liver diseases. This could have a positive impact on future therapeutic strategies.

## 1. Gut Microbiota

The gut microbiota (GM) is a diverse ecosystem that consists of bacteria, protozoa, archaea, fungi, and viruses, which exist in a specific symbiosis between each other and the human body as well. Currently, it is well known that GM plays relevant roles in physiological and pathological conditions of human health, taking part in digestion, vitamin B synthesis, immunomodulation, and promotion of angiogenesis and nerve function. In addition, it is unavoidable that the GM has an impact on pathogenesis of gastrointestinal, hepatic, respiratory, cardiovascular, endocrine, and many other disorders, arising as “a new virtual metabolic organ” [1].

The GM colonizes human intestinal tract, which accounts for more than 100 trillion bacteria, and has a complex genome of 150-fold more genes than the human genome [2]. The majority of gut microorganisms cannot be cultured using standard techniques, so the development of culture independent molecular methods based on sequencing of the phylogenetic marker—16S/18S ribosomal RNA offer better insight in the GM structure. The GM is essentially ecomposed of the five phyla-*Firmicutes* (79.4%) (*Ruminococcus*, *Clostridium*, and *Eubacteria*), *Bacteroidetes* (16.9%) (*Porphyromonas*, *Prevotella*), *Actinobacteria* (2.5%) (*Bifidobacterium*), *Proteobacteria* (1%), and *Verrumicrobia* (0.1%) [3]. *Lactobacilli*, *Streptococci*, and *Escherichia coli* are found in small numbers in the gut. Different genetic and environmental factors influence the GM composition. For example, children born by natural childbirth inherit about 40% of the mother’s intestinal flora, while GM composition is very different after the caesarean section. During the first two years of life, the diet is the most prominent factor that determines GM. Later in life, GM composition depends on age, diet, medications, and the environment.

Studies published in the last decade confirmed that the GM is implicated in the pathogenesis of various diseases, such as cancer and autism, depression, *Clostridium difficile* infection, inflammatory bowel disease, irritably bowel syndrome, colorectal carcinoma, infectious and non-infectious chronic liver diseases, obesity, diabetes mellitus type 2, atherosclerosis, and chronic kidney diseases [4,5,6,7,8,9]. In the present review, the important role of GM in the pathogenesis of most common liver diseases was discussed.

## 2. Gut-Liver Axis

GM as a “virtual metabolic organ” makes axis with a number of extraintestinal organs, such as kidneys, brain, cardiovascular, and the bone system, but the gut-liver axis attracts increased attention in recent years [10]. The gut-liver axis is a consequence of a close anatomical and functional, bidirectional interaction of the gastrointestinal tract and liver, primarily through a portal circulation. The symbiotic relationship between the GM and the liver is regulated and stabilized by a complex network of interactions, that encompass metabolic, immune, and neuroendocrine crosstalk between them [4]. The tight junctions (TJ) within the gut epithelium represent a natural barrier to bacteria and their metabolic products [11]. Antigens (Ag) (originating either from pathogenic micro-organisms or from food) that pass through these connections, are recognized by dendritic cells, or activate the adaptive immune system by modulating the T cell response. Minimal concentrations of pathogen-associated molecular patterns (PAMPs), such as lipopolysaccharides (LPS), peptidoglycans, and flagelin, activate the nuclear factor kappa B (NFKβ) through toll-like receptors (TLRs) and nod-like receptors (NLRs), which leads to the production of inflammatory cytokines and chemokines that enter portal circulation. In addition to hepatocyte damage, PAMPs can activate stellate cells involved in fibrosis promotion and progression, while Kupffer cells are even more responsive to LPS than hepatocytes [12]. Since the gut-liver axis affects the pathogenesis of liver diseases, it is an important focus of current clinical research (Scheme 1).

## 3. Disbyosis and Liver Diseases

The gut-liver axis has an impact on pathogenesis of numerous chronic liver diseases such as chronic hepatitis B (CHB), chronic hepatitis C (CHC), alcoholic liver disease (ALD), non-alcoholic fatty liver disease (NAFLD), non-alcoholic steatohepatitis (NASH), development of liver cirrhosis, and hepatocellular carcinoma (HCC) (Table 1).

In general, an increased intestinal permeability and bacterial translocation could enable microbial metabolites to reach the liver, which would impair the bile acid (BA) metabolism and promote gut dysmotility and systemic inflammation. All these conditions could induce gut dysbiosis, which, in turn, further increases liver damage. It has been observed that the stage of liver injury correlates closely with the severity of gut dysbiosis [23]. Alterations in the fecal bacterial flora are described by changes in the composition of the dominant *Bacteroidetes* and *Firmicutes* phyla, including *Ruminococcaceae*, *Lachnospiraceae*, and *Clostridiales*, which produce short-chain fatty acids (SCFA) that are an energy source for the intestinal epithelium’s cells, but can also regulate secondary BA metabolism and induce a regulatory immune process and IgA production.

### 3.1. Hepatitis B Virus (HBV) Infection

CHB is an important health issue worldwide. Acute HBV infection leads to CHB in just 5% of adult patients, while the proportion is quite different in children, since more than 90% of exposed neonates and 30% to 50% of children aged 1 to 5 years fail in HBV clearance. Liver injury is mediated by HBV induced immune response. TLRs play an important role in the production of interferons and proinflammatory cytokines and immune cells recruitment in order to suppress viral replication. It has been established that age-specific seroclearance depends not only on the maturity of the immune system, but on the GM stability as well [24]. Involvement of GM in HBV clearance was demonstrated in animal models. Chou et al. showed that adult mice with mature GM managed to clear HBV after six weeks of infection, which is the opposite among young mice without GM, who remained HBV positive. The fact that adult mice failed to clear HBV after gut sterilization by antibiotics (6 to 12 weeks), emphasizes the GM significance in anti-HBV immunity [25]. It also implies new therapeutic strategy for patients with HBV infection [26]. In fact, the transplantation of fecal microbiota (FMT), in addition to standard antivirals, has been shown to be effective in HBeAg clearance [27].

Compositional and structural changes of GM have been detected in patients with CHB and liver cirrhosis. These patients have a decreased ratio of *Bifidobacteriaceae*/*Enterobacteriaceae* (B/E), based on low levels of *Bifidobacteria* and *Lactobacillus*, and high levels of *Enterococcus* and *Enterobacteriacea*. In addition, gut permeability is altered when accompanied with bacterial translocation and the presence of endotoxins in the portal vein, which leads to increased TLS/NLR activation in the liver with consequential cytokine production and occurrence of liver lesions, progression of fibrosis, and development of cirrhosis and HCC [2,13]. Wei et al. have demonstrated that GM of patients with HBV-related cirrhosis contained lower levels of *Bacteroidetes* (4% vs. 53%) and increased levels of *Proteobacteria* (43% vs. 4%) compared to the heathy group [14]. In an other study, patients with alcohol-related and HBV-related cirrhosis showed decreased GM diversity, compared to healthy individuals, with a predominance of *Enterobacteriaceae* and *Streptococcaceae* [28].

### 3.2. Hepatitis C Virus (HCV) Infection

CHC is a global health problem that leads to progressive liver fibrosis and the cirrhosis development in 20% to 30% of untreated patients after 20 to 30 years. It has been estimated that 1% to 4% of these patients develop HCC each year [29]. GM has been rarely analyzed in patients with HCV infection. According to published data, the GM found in HCV patients shows lower microbial diversity in comparison to those in healthy controls [15,28,30]. CHC could alter microbiota composition through IgA produced by HCV infected gastric B-lymphocyte. GM found in Egyptian patients with CHC contains more *Prevotella* and *Faecalibacterium* and less *Acinetobacter*, *Veillonella*, and *Phascolarctobacterium* than healthy individuals. In the study of Aly et al., *Bifidobacterium* was detected only in GM of the healthy group, posing the possible new role of *Prevotella*/*Faecalibacterium* vs. *Bifidobacterium* ratio as a biomarker for CHC and fibrosis progression [15]. Disease progression could bring more profound changes in CHC patients’ GM. Therefore, according to Heidrich et al., decreased diversity was more pronounced in HCV patients’ with established cirrhosis than in those with less advanced CHC [31]. Liver cirrhosis *per se* could be an independent risk factor for dysbiosis regardless of the HCV viral load. This hypothesis is in agreement with a study performed by Bajaj et al. who found that patients with HCV cirrhosis have gut dysbiosis regardless of long-term HCV eradication. A sustained virological response (SVR) did not improve gut dysbiosis in patients with HCV cirrhosis, due to refractory systemic inflammation and endotoxemia in these individuals [32]. Bacterial translocation was described in patients with CHC and together with increased intestinal permeability (“leaky gut”), it poses a well-established milieu with TLR/NLR activation and expression of pro-inflammatory cytokine genes, especially in those with cirrhosis [33].

### 3.3. Alcoholic Liver Disease

Alcohol abuse is a prominent cause of liver damage worldwide. GM is recognized as a key player in the severity of liver injury in ALD, in addition to the quantity of consumed alcohol and genetic predisposition (patatin like phospholipase domain containing 3 (PNPLA3), Transmembrane 6 superfamily 2 human gene (TM6SF2), membrane bound O-acyltransferase domain containing 7 (MBOAT7), and solute carrier family 38 member 4 (SLC38A4) etc.) [34]. Alcohol consupmtion leads to small and large intestine overgrowth and modulation of GM composition in both animals and humans [35]. Alcohol and its degradation products disrupt epithelial TJ leading to increased intestinal permeability and inflammation [34]. Gut-derived PAMPs (e.g., endotoxin) are increased after heavy alcohol intake [36]. Ethanol consumption alters the GM composition through SCFAs modulation. Intestinal levels of SCFAs are lower after alcohol consumption with the exception of increases in acetic acid levels, which is the metabolite of ethanol [37]. Alcoholic abuse was shown to be associated with decreased levels of butyrate-producing Clostridiales species order and increased levels of pro-inflammatory Enterobacteriaceae. In those with established cirrhosis, multiple members of the Bacteroidales order were depleted with a rise of taxa normally inhabiting the oral cavity [19].

### 3.4. Non-Alcoholic Fatty Liver Disease and Non-Alcoholic Steatohepatitis

NAFLD is one of the most important causes of liver disease worldwide, with global prevalence of 25%. NAFLD is one of the top risk factors for HCC and is predicted to become the most common indication for liver transplantation [38]. NAFLD is a consequence of triglyceride accumulation in the hepatocytes and is considered to be the hepatic manifestation of obesity and the metabolic syndrome [39]. About 20% of patients with NAFLD develop NASH, which is a chronic hepatic inflammation that can progress to cirrhosis, end stage liver disease (ESLD), and HCC. Pathogenesis of NASH is not yet fully elucidated, but it is described as a “two hit” phenomenon. The primary event is lipid accumulation with alterations of lipid homeostasis associated with obesity, insulin resistance, and adipokine abnormalities. The second “hit” is a combination of oxidative stress, lipid peroxidation, mitochondrial dysfunction, BA toxicity, cytokine-mediated recruitment, and retention of inflammatory cells [40]. Obesity is associated with dysbiotic gut microbiota, with decreased diversity and an increased *Firmicutes*/*Bacteroidetes* ratio [20]. A similar *Firmicutes*/*Bacteroidetes* ratio was found in diabetes patients as well. Endogenous ethanol is constantly produced by microbiota, regardless of oral alcohol intake, especially in those with a sugar-rich diet. Increased ethanol production by microbiota in obese humans and mice leads to the activation of TLRs in the liver, cytokine production, and alters the BA profile. Endogenous ethanol serum concentrations are significantly higher in patients with NASH compared to obese or healthy controls [41]. Gut dysbiosis in patients with NAFLD/NASH promotes insulin resistance, *de novo* lipogenesis in liver, and also increases intestinal permeability, which promotes chronic PAMPs exposure and oxidative stress caused by increased endogenous ethanol [21]. Endotoxin/TLR4 signalling contributes to the development of fibrosis and progression to cirrhosis through hepatic stellate cell activation. The GM plays a critical role in the conservation of the mainstream BA pool, which transforms BA to several metabolites by oxidation/epimerization, deconjugation, esterification, 7-dehydroxylation, and desulfatation. Changes in any of these modulations is a cause of disease. GM dysbiosis in NAFLD could affect the conversion of primary bile acids into secondary bile acids [42]. It has been observed that the bacteria able to make this transformation are decreased in the NAFLD cirrhosis fecal samples. In particular, there is a higher level of *Enterobacteriaceae* (that could be potentially pathogenic) with lower *Ruminococcaceae*, *Lachnospiraceae*, and *Blautia* (with a 7α-dehydroxylating activity) abundances.

Currently, the roles of BA and nuclear receptors are also strongly considered. The BA primarly synthesized in the liver are secreted to the gallbladder and then released into the duodenum following food ingestion. In the gut, the size and composition of the BA pool can be modified by GM via the biotransformation of primary into secondary BA. The BA contributes to the emulsification and fat solubilization but also activates the expression of a nuclear bile acid receptor FXR (farnesoid X receptor) and a membrane G protein-coupled receptor TGR5. The reduction of the secondary BA synthesis attributed to GM dysbiosis lowers the activation of nuclear receptors FXR and TGR5 in the ileum, which leads to retained bile salts, perpetuation of gut permeability, small bowel translocation, and bacterial overgrowth, which contributes to liver disease [43]. The FXR plays a key role in mediating the crosstalk between the host and GM, especially through the modulation of enterohepatic BA circulation. FXR exerts bile-acid regulatory effects via a tissue-specific mechanism. [44]. In detail, in the liver, FXR induces the expression of the small heterodimer partner (SHP), which inhibits CYP7A1 (Cholesterol 7 alpha-hydroxylase) expression, while, in the intestine, FXR increases the levels of circulating fibroblast growth factor 19 (FGF19), which decrease the expression of CYP7A1 and cytochrome P450 12a-hydroxylase B1 (CYP8B1). Therefore, this leads to the inhibition of BA synthesis [45].

FXR activation has been known to reduce triglyceride levels suppressing the synthesis and uptake of the fatty acids in the liver [46]. In addition, the FXR roles in decreasing inflammation have been emerging [47]. Lastly, related to glucide metabolism, FXR reduces insulin resistance, gluconeogenesis, increases glycogenesis, and, therefore, decreases the blood glucose amount. A more fine elucidation of FXR functions in liver and intestine needs further research. Furthermore, mice models showed that FXR activation, induced by BA products (converted by GM) might protect against bacterial overgrowth, gut permeability, and small bowel translocation [48]. The degree of FXR activation could be regulated by GM dysbiosis, inducing BA alteration and, hence, liver disease secondary to retained bile salts and a leaky gut. Bacteria, translocated from the gut, may additionally decrease FXR activation within hepatocytes, which leads to decreased BSEP (bile salt export pum) activity.

The bile acid-activated FXR and TGR5 has a protective role in liver disease progression, which means the activation of both receptors has been proposed as therapy. Many activators of FXR and TGR5 have been developed from bile acid analogues, which are able to decrease hepatic steatosis and inflammation [49]. FXR agonists such as obeticholic acid have already been recognized as a new therapeutic venue for NASH and cholestatic diseases. NASH patients exhibit increased fecal primary BAs, primary/secondary BAs fecal ratio, and plasma and hepatic BAs concentrations [50]. This cytotoxicity can lead to NASH progression and finally to liver cirrhosis.

### 3.5. Hepatic Encephalopathy (HE) and Spontaneous Bacterial Peritonitis (SBP)

In all patients with ESLD, regardless of its etiology, HE and SBP commonly occur. HE is considered to be a consequence of high ammonia level, but GM and bacteria’s products, such as amino acid metabolites (indoles, oxindoles) and endotoxins, are also involved in HE development. The connection between HE and GM is the production of ammonia and endotoxins by urease-producing bacteria, such as Klebsiella and Proteus, which are present in GM [18]. It has been shown that microbiota of the sigmoid colon in liver cirrhosis patients differs in those with HE when compared with patients without HE [51]. Gut dysbiosis in liver cirrhosis also contributes to the development of Spontaneous Bacterial Peritonitis (SBP) through damaged intestinal barrier and a higher degree of microbial translocation [52].

### 3.6. Hepatocellular Carcinoma

HCC is the most common primary malignancy in adults with chronic liver disease and liver cirrhosis [53]. HBV and HCV infection, alcohol abuse, and dietary aflatoxin are major risk factors for the development of HCC. Although HBV and HCV account for 80% to 90% of overall HCC, regarding the obesity pandemic, an emergance of potent direct acting antivirals for HCV and worldwide available vaccine for HBV, it can be expected that HCC epidemiology will change in the future [54]. An alarming rise in the incidence of NAFLD and NASH was accompanied by an increased development of NASH-related HCC incidence [55].

All previously mentioned mechanisms, leaky gut, endotoxemia, TLR, dysbiosis, and immunomodulation promote the development of HCC [56]. The gastrointestinal tract contributes to homeostasis by maintaining an intact barrier against LPS and intestinal bacteria. In the case of increased intestinal permeability, bacterial translocation and LPS accumulation will lead to intestinal bacterial overgrowth and changes of GM composition. In patients with chronic liver diseases/cirrhosis, detoxification, degradation, and clearance of LPS and other bacterial products is compromised [57]. Altered microbiota is generally presented in HCC patients [58]. HCC patients have been reported to have high levels of *Escherichia coli* and other gram-negative bacteria in GM, which are associated to increased LPS serum levels [16]. *Oribacterium* and *Fusobacterium* are the most commonly isolated bacteria from tongue swab of HCC patients. On the other side, GM of HCC patients contains reduced levels of *Lactobacillus* spp., *Bifidobacterium* spp., and *Enterococcus* spp. [17]. Unlike merely microbial species, it was shown that microbial metabolism, iron transport, and energy-producing system significantly differ between GM of HCC patients and healthy control [58]. TLR4 expressed by activated stellate cells react to low LPS concentrations, which ensures the development of fibrosis and cirrhosis. The significance of GM and TLR4 activation in hepatocarcinogenesis have been studied in an animal model. It has been demonstrated that GM and TLR4 activation promote HCC development by increased cell proliferation and suppression of apoptosis [59].

## 4. Current Perspectives on the Therapeutic Options

Dietary modification is the focus for many studies on GM modification, but results are not encouraging due to poor compliance [11]. Different treatments options have been developed in the last decade in the attempt to modify pathogenesis factors involved in GM-liver axis, but results are still unsatisfactory [60]. Antibiotics, probiotics, prebiotics, and symbiotics are gaining increasing importance as a treatment option for GM manipulation and its impact on different liver diseases (Table 2).

### 4.1. Antibiotics, Probiotics, Prebiotics, and Symbiotics

Positive effects of antibiotics on small-intestinal-bacterial-overgrowth related liver damage was shown in the recent literature, which confirms the relationship between GM and liver diseases [70]. Probiotics are micro-organisms with a healthy impact to the human, while pre-biotics are food ingredients, which can not be digested and, on that way, help gut peristaltic and selectively stimulate growth of intestinal bacteria. Some prebiotic food products, like pectin, have been shown to prevent liver injury in rodent models by restoring the levels of *Bacteroides* and appears to be a promising therapeutic agent [71]. New term “symbiotics” represent combinations of a probiotic and prebiotic. From this group, the most potential for being used as potential treatment of chronic liver diseases have probiotics [11,70]. Probiotics have a significant influence on the gut-liver axis, including the immunomodulatory and anti-inflammatory role on intestinal microflora and intestinal barrier function, but also metabolic effects on non-gastrointestinal organs and systems.

Not all probiotic species have the same impact on GM modulation. For example, different strains of *Lactobacillus* spp. are associated with divergent obesity consequences, pro-obesity and anti-obesity [72]. A study on murine diet-induced obesity showed the opposite metabolic outcomes after different targeted GM manipulation with vancomycin vs. probiotic strain *Lactobacillus salivarius*, but microbiota changes were similar. On the other side, a recent human study investigating the effect of the *Lactobacillus salivarius* Ls-33 on a series of inflammatory biomarkers in obese adolescents could not detect any significant impact on the metabolic syndrome [73,74]. These results suggest that no generalization could be done. Use of probiotics and antimicrobials in chronic liver diseases should be determined in accordance to specific gut-liver axis pathology. Although there are a number of findings from animal and human studies, this topic requires more case-control prospective studies with a significant number of patients.

### 4.2. Fecal Microbiota Transfer (FMT)

Next to probiotics and prebiotics, fecal microbiota transfer attracts a lot of attention currently as a possible option for GM editing. FMT is the introduction of a fecal suspension derived from a healthy donor into the intestinal tract of the patient. Animal studies revealed altered GM composition after FMT in alcohol-sensitive mice and prevented alcohol-induced liver lesions [71]. Wang et al. showed potential protective effects of FMT in the improvement of motor activity in a rat model of HE. In addition, FMT had a superior impact on alteration of the intestinal mucosal barrier function, in comparison to probiotics [75]. A different study with the rat model of HE found that rectal administration of a *Lactobacillus* species suppressed bacterial cell proliferation [76]. Regarding the clinical trial, a pilot study was performed in eight male patients with severe alcoholic hepatitis (SAH) and compared outcomes with historical controls. Researchers found that one week of FMT was effective and safe in SAH patients and improved the values of liver disease severity and survival at one year [77]. In addition, an open label randomized clinical trial was performed in 20 male patients with cirrhosis and recurrent HE despite treatment with lactulose and rifaximin, comparing FMT vs. standard of care [78]. The result showed that FMT from a rationally selected donor (chosen for having the highest abundance of *Lachospiraceae* and *Ruminococcaceae* in a universal stool bank) improved cognition, reduced hospitalizations, and ameliorated the dysbiosis condition in cirrhosis with recurrent HE. However, it is not yet known what consequences FMT may excert on dysbiosis in the upper gastrointestinal tract, which may be more involved in liver disease pathogenesis. Further clinical trials benefiting FMT for management of chronic liver diseases are ongoing. It is important to highlight that FMT is a promising therapy to restore a healthy microbiota. However, its safety profile among vulnerable liver disease patients that are usually immunosuppressed and, so at risk of bacteremia, due to bacterial translocation, is unknown.

### 4.3. NAFLD

Several pharmacologic treatments for NAFLD exist, but a standard treatment is not yet established [79,80]. Recent animal model-studies on NAFLD and HE reported beneficial therapeutic effects of probiotics [81,82,83]. In mice models, probiotics had positive effects on oxidative and inflammatory liver damage mediated by c-Jun N-terminal kinase (JNK) and NF-KB correlated with TNF-α regulation and insulin resistance. The results showed improvement of histological findings including reduced fat deposits and liver damage, with decreased serum alanine aminotransferase (ALT) levels [81,83,84]. Apart from animal studies, several clinical trials of probiotics administered to patients with NAFLD have been reported. The mixture of *Streptococcus thermophilus*, *Bifidobacterium breve*, *Bifidobacterium longum*, *Bifidobacterium infantis*, *Lactobacillus acidophilus*, *Lactobacillus plantarum*, *Lactobacillus paracasei*, and *Lactobacillus bulgaricus* has joint name “VSL #3” and represent the most-studied probiotic [61]. A meta analysis of effects of probiotic on NAFLD/NASH showed that probiotic therapies can reduce liver aminotransferases, total-cholesterol, TNF-α, and improve insulin resistance in NAFLD/NASH patients [62]. Administration of non-complete VSL #3 (only *Lactobacillus bulgaricus* and *Streptococcus thermophilus*) decreased ALT and aspartate aminotransferase (AST) levels, but had no impact on cardiometabolic risk factors [85]. Ma et al. found that the probiotic therapy significantly decreased the levels of ALT, AST, total-cholesterol, high-density lipoprotein (HDL), and TNF-α in the serum, and the homeostasis model assessment of insulin resistance [62]. Comparing single strains, *Bifidobacterium longum* showed more positive effects than *Lactobacillus acidophilus* in NAFLD treatment and these beneficial effects correlated to GM modifications [61]. It was demonstrated that GM modulation by linoleic acid and *Bifidobacterium* resulted in an increase of conjugated linoleic acid (CLA) and altered fatty acid composition of liver [86]. Consequently, by producing CLA as a microbial metabolite, VSL #3 correlates with NAFLD’s improvement [86]. In additional studies (of Malaguarnera et al. and Tanock et al.), probiotics were used with prebiotics in NAFLD patients, and the combination significantly reduced the AST and ALT levels and liver steatosis [87,88].

Prebiotics also hold promise for the NAFLD management. In animal models, the administration of prebiotics could lead to reduced liver inflammation in obese mice through a glucagon-like peptide-2-dependent effect on the gut barrier [89]. There is convincing evidence from short-term high quality human trials supporting the use of dietary prebiotics as a potential therapeutic intervention for NAFLD. However, further studies are needed to correlate these findings with changes in GM [90]. Although most of the conducted studies have limitations including small sample sizes and a lack of data about the patients’ diets and life style, the treatments with probiotics and prebiotics for patients with NAFLD are promising [91].

Moreover, a recent meta-analysis study demonstrated that, in addition to probiotics and prebiotics, ω-3 supplementation could have effects on the gut microbiota, which improves NAFLD [92]. It has been shown that n-3polyunsaturated fatty acids (n-3 PUFAs) decrease *Akkermansia*, *Epsilon proteobacteria*, *Bacteroides*, and increase *Clostridia* [93]. Lastly, the n-3 PUFAs administration has been related to an increment of butyrate-producing bacteria [94].

### 4.4. Cirrhosis

Qin at el. showed reduced *Bacteroidetes* and *Firmicutes* levels and increased *Streptococcus* spp. and *Veillonella* spp. levels in GM by metagenomics analyses in cirrhotic patients, compared to a healthy population [22]. *Streptococcus* spp. and *Veillonella* spp. are bacteria regularly living in oral cavity, which implicates that an increased number of oral bacteria in GM may be related to cirrhosis development [16]. It was reported that the gut microbiome plays a role in the CHB progression to severe liver failure, including inflammation and pathogenic metabolic accumulation [75]. The connection between GM and cirrhosis was recently confirmed in the study of Bajaj et al., which showed a significant improvement of diversity and symbiosis of GM after liver transplantation in a patient with severe cirrhosis [95]. In the recent international cirrhosis cohort study, Bajaj et al. showed that diverse GM achieved by diet is associated with a lower risk for cirrhosis progression and reduced risk of hospitalization [63]. Although the functions of these bacteria in the pathogenesis and complications of liver cirrhosis is not yet clear, these findings give hope for a new therapeutic strategy against liver cirrhosis by focusing on the GM modulation.

### 4.5. Hepatic Encephalopathy

Probiotics inhibit the activity of bacterial ureases, modulate intestinal pH values, and reduce ammonia absorption in HE [96]. Administration of probiotics and prebiotics could improve HE by altering GM [18]. It is already shown that *Lactobacillus*, *Bifidobacterium*, non-pathogenic strains of *Escherichia coli*, *Clostridium butyricum*, *Streptococcus salivarius*, and *Saccharomyces boulardii*, and VSL #3 altered GM composition and improve HE [97]. A number of conducted clinical trials indicate that probiotics could be helpful in overt HE [65,66,67,68,69]. In a systematic review of McGee et al., it was found that patients treated with probiotics appeared to have reduced plasma ammonia concentrations compared to patients treated with placebo or no intervention. On the other side, treatment without probiotics and synbiotics did not significantly alter mortality and quality of life [65]. The possible explanation is that the sample of study groups was small in the reviewed control trials, as well as quantities of the probiotics were not uniform. Use of probiotics for the secondary HE prophylaxis was studied by Agrawal et al. Although not statistically significant, the occurrence rate of recurrent HE was lower in patients who received probiotics or lactulose compared to those without treatment (34% and 27%, respectively, vs. 57%) [67]. In another prospective study by Lunia et al., it was shown that probiotics could be effective in preventing overt HE. Patients with liver cirrhosis who had not experienced overt HE and who received the probiotics were less likely to develop overt HE compared to controls (1.2% vs. 19%) [67]. Among the various probiotics, the most efficacious species for HE appeared to be *Lactobacilli* and *Bifidobacteria* [97]. The most studied probiotic is the *Lactobacillus* GG AT strain 53103 (LGG). Safety and tolerability of administrated therapy is especially important for the patients with cirrhosis and HE. It is shown that LGG is safe and well tolerated in patients with cirrhosis and could cause reductions of the endotoxemia and TNF-α production [98]. However, due to a specific clinical course, randomized controlled trials are needed before probiotics can be routinely used in patients with HE.

### 4.6. HCC

The mouse model has shown that the obesity-induced intestinal microbial dysbiosis can lead to HCC [99]. Reports of the therapeutic prevention of HCC using probiotics are limited to aflatoxin-induced HCC. Findings of a clinical and animal model studies suggested that probiotics can contribute to the inhibition of aflatoxin B-induced hepatocarcinogenesis, restore intestinal dysbiosis, reduce LPS levels, and decrease the tumor size [59,100]. An animal study regarding the potential of Lactobacillus plantarum, isolated from Chinese traditional fermented foods, in reducing the toxicity of aflatoxin B1, showed that L. plantarum C88 treatment increased fecal aflatoxin B1 excretion and regulated defense system’s deficit of antioxidant in the mice model [100]. Lastly, a study with an HCC rat model showed that probiotic fermented milk and chlorophyllin slow down tumor growth and volume for 40%, by reducing expressions of c-myc, bcl-2, cyclin D1, and rasp-21. After treatment with probiotics, mice had high levels of *Prevotella* and *Oscillibacter* in their fecal microbiota [64]. Dapito et al. also found that gut sterilization and TLR4 inactivation reduce HCC by 80% to 90% and could serve as potential HCC prevention strategies [59].

## 5. Conclusions

Knowledge regarding the gut-liver axis has improved in the last decade. It is confirmed that gut microbiota has a strong relationship with liver and plays a significant role in the pathogenesis of chronic liver diseases, including NAFLD, fibrosis progression, CHB, cirrhosis, HE, HCC, etc. However, the mechanism of this connection in different liver diseases is still unclear. Our knowledge about the clinical significance of probiotics’ use in liver disease is starting to take shape. Although clinical and experimental studies confirmed the therapeutic potential of probiotics in chronic liver diseases, data on safety assessments, and impact on microbiota-host interactions are missing. Recently, many animal studies tried to reveal these gaps, but differences in physiology and variations in the molecular targets between mice and humans can lead to translational limitations. Large prospective controlled studies with a standardized dose of probiotics therapy and duration, liver biopsy, and patients’ follow-up appointments are required to confirm these encouraging results.

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
