# Peer review of "Gut-Liver Axis, Gut Microbiota, and Its Modulation in the Management of Liver Diseases: A Review of the Literature"

_ijms, 2019, doi:10.3390/ijms20020395_

Reviewer 1 Report

The manuscript summarizes the role of gut microbiota dysbiosis in the pathogenesis of various liver diseases, and discusses potential therapies based on manipulation of the gut microbiota. The review covers an interesting and clinical relevant topic. Some pitfalls however impairs the overall significance. The following are the comments and critique:

1.      Alcoholic liver disease is one the main causes of chronic liver diseases and the third most common cause of liver transplants. Researchers in the field have reported alterations of microbiota composition, changes in gut microbiota-related metabolites, correlations between microbiota dysbiosis and disease progression, and the treatments in alcoholic liver disease. The manuscript only mentioned one work in “4.2. Fecal microbiota transfer” (ref. 58). I would suggest that the authors have another section describing microbiota and alcoholic liver disease.

2.      The layout of Section 4 “Current perspectives on the therapeutic options” is suggested to be re-organized. Section 4.1 and 4.2 are about different methods, including antibiotics, probiotics, FMT, etc., whereas Sections 4.3 to 4.6 are about various diseases. And under each disease, treatments such as probiotics, prebiotics are repeatedly discussed. Please keep one order of layout.

3.      Wherever studies and results are reported, the original research study should be cited. For example, Line 172 cites ref. 37 which is a review. The original study should be Hepatology 2013; 57:601-609.

4.      Line 60-61, do the authors mean “majority of gut microorganisms could not be analyzed”? The 16S/18S rRNA sequencing method could not be used to “culture” bacteria.

5.      Proof reading is suggested. For example, Line 45, it should be “gut-liver”. Line 325, it should be “showed” instead of “shown”.

Author Response

International Journal in Medical Sciences

11.01.2019.

Manuscript No. ijms-423157

“Gut-liver axis, gut microbiota and its modulation in the management of liver diseases: a review of the literature”

COVER LETTER AND RESPONSE TO REVIEWER’S COMMENTS

Dear Editor,

Thank you very much for support and considering our manuscript entitled “Gut-liver axis, gut microbiota and its modulation in the management of liver diseases: a review of the literature” for the publication in International Journal in Medical Sciences and for more than useful reviewers’ comments. We would like to thank you and the reviewers for your valuable time, feedback and comments with regards to our submission. We read carefully all reviewers comments and used this opportunity to submit response to Editor and reviewers. We have now addressed the issues raised by yourself and each reviewer and highlighted all changes throughout our manuscript for your convenience. The paper was re-checked for language errors by native English speaker and all technical mistakes are corrected throughout the article text and corrections are highlighted.

Reviewer 1 comments:

Reviewer 1: The manuscript summarizes the role of gut microbiota dysbiosis in the pathogenesis of various liver diseases, and discusses potential therapies based on manipulation of the gut microbiota. The review covers an interesting and clinical relevant topic. Some pitfalls however impairs the overall significance. The following are the comments and critique:

Reviewer 1:Alcoholic liver disease is one the main causes of chronic liver diseases and the third most common cause of liver transplants. Researchers in the field have reported alterations of microbiota composition, changes in gut microbiota-related metabolites, correlations between microbiota dysbiosis and disease progression, and the treatments in alcoholic liver disease. The manuscript only mentioned one work in “4.2. Fecal microbiota transfer” (ref. 58). I would suggest that the authors have another section describing microbiota and alcoholic liver disease.

Author: Dear reviewer, thank you very much for support and for more than useful comments that helps us a lot with manuscript improvement. We respected them all and edited the manuscript in accordance with your suggestions. We agree that “alcoholic liver disease” is inevitable when we discuss about liver diseases, so we include this diagnosis in the review (please see lines: 181-194)

Reviewer 1: The layout of Section 4 “Current perspectives on the therapeutic options” is suggested to be re-organized. Section 4.1 and 4.2 are about different methods, including antibiotics, probiotics, FMT, etc., whereas Sections 4.3 to 4.6 are about various diseases. And under each disease, treatments such as probiotics, prebiotics are repeatedly discussed. Please keep one order of layout.

Author: In the section 4 “Current perspectives on the therapeutic options”, we gave a short introduction about potential therapeutic options in general in the term “What they are”, and then, we discussed their use in different liver diseases, one by one. The idea of this review is to discuss relationship of individual liver diseases and gut microbiota, and their potential microbiota-related treatment options. We gave very short explanation what are potential therapeutic options here (antibiotics, probiotics, FMT etc) and then went into details one by one disease. It is more like “Introduction” in this section. We reorganized repeated facts, but in general, we left this introduction of section 4. I hope explanation will give reviewer better overview what was idea and why we wrote these sentences, but if reviewer prefers, we will delete them.

Reviewer 1:  Wherever studies and results are reported, the original research study should be cited. For example, Line 172 cites ref. 37 which is a review. The original study should be Hepatology 2013; 57:601-609.

Author: Thank you for the suggestion and pointing to mistake. We corrected the requested reference in accordance to the cited study.

Reviewer 1:   Line 60-61, do the authors mean “majority of gut microorganisms could not be analyzed”? The 16S/18S rRNA sequencing method could not be used to “culture” bacteria.

Author: In the non-reviewed form, this sentence was confusing. We corrected it with aim to make it clearer. Now, the sentence is: “Majority of gut microorganisms  can not be cultured using standard techniques, so the development of culture independent molecular methods based on sequencing of phylogenetic marker – 16S/18S ribosomal RNA offer better insight in GM structure”.

Reviewer 1:  Proof reading is suggested. For example, Line 45, it should be “gut-liver”. Line 325, it should be “showed” instead of “shown”.

Author: The paper was re-checked for language errors by native English speaker and all technical mistakes are corrected throughout the article text and highlighted in yellow.

Reviewer 2 Report

Authors of this Review for IJMS aimed to summarize aspects linking GLA players and a group of unrelated liver diseases. 

Authors should acknowledge that the story is much more complex, and for each of the conditions they have described there are several  pieces of oceanic information. E.g., see how others tried to summarize the roles of the many players involved in GLA/microbiota and NAFLD:

1: Poeta M, et al . Gut-Liver Axis Derangement in Non-Alcoholic Fatty Liver Disease. Children (Basel). 2017 Aug 2;4(8).

2: Clemente MG, et al. Pediatric non-alcoholic fatty liver  disease: Recent solutions, unresolved issues, and future research directions.

World J Gastroenterol. 2016 Sep 28;22(36):8078-93

3. Nobili V,  et al. . Fighting Fatty Liver Diseases with Nutritional Interventions, Probiotics, Symbiotics, and Fecal Microbiota Transplantation (FMT). Adv Exp Med Biol. 2018 Dec 22.

On the other hand the same treatment done for NAFLD should be reserved to their HBV, HCV, HCC, HE sections  and so on. We understand that this might need a book. Therefore, in my opinion, Authors should focus better which is their MS's target. Once decided, they should be extremely precise and avoid putting crowds of information together.

Said that, the topic chosen is not new. Several innovative reviews capable of giving unifying hypotheses  have been published and continue to be published on that issue. As submitted  in its present form the Authors' MS deals with a relevant topic, but then it appears  a mix of drops in the ocean, not inter/intra - linked each-other,  which appear to have been selected without taking into account the endless explosion of very specific acquisitions in the field.

The conclusion the readers can draw from this MS  risks  therefore to be unhelpful to experts and to non-experts as well.

Specific issues:

A. AUTHORSHIP: In my personal opinion 12 Authors are too many for a review. Authors should give some stronger reason than that shown in the dedicated section of the MS.

B. TEXT READIBILITY: The text is difficult to read.

1.     Please indicate with headings/sub headings when you change topic

2.     A few Summarizing Tables might be OK for improving the readibility.

3.     Similarly, a Summarizing synoptic picture of GLA and liver diseases might help to follow better the issues which have been or are going to be treated in the text;   

4. PNALD, IBD, Cystic Fibrosis, PSC should be included among the liver diseases of interest:  e.g. see as others have done for celiac disease/GLA and liver : Marciano et al.  Dig Liver Dis. 201Feb;48(2):112-9.

C. "CHB is an important health issue worldwide …": The subsequent text from line 108 to 115 is too detailed and appears out of the MS scope. 

D. Ethanol: Authors please clearly specify that it is Endogenous EtOH

E. FXR: Bile acid synthesis, negative feedback inhibition through FXR, hepatic and ileal FXR  expression;  FXR  role in lipid and glucide metabolism;  dysbiosis elated abnormal bile acid modifications steps, change of FXR activation and signaling as causes of  cholestasis and leaky gut dependent liver changes should be better clarified  (see the enlightening MS of Leung DH, Yimlamai D. The intestinal microbiome and paediatric liver disease. Lancet Gastroenterol Hepatol. 2017 Jun;2(6):446-455.)

Author Response

International Journal in Medical Sciences

11.01.2019.

Manuscript No. ijms-423157

“Gut-liver axis, gut microbiota and its modulation in the management of liver diseases: a review of the literature”

COVER LETTER AND RESPONSE TO REVIEWER’S COMMENT

Dear Editor,

Thank you very much for support and considering our manuscript entitled “Gut-liver axis, gut microbiota and its modulation in the management of liver diseases: a review of the literature” for the publication in International Journal in Medical Sciences and for more than useful reviewers’ comments. We would like to thank you and the reviewers for your valuable time, feedback and comments with regards to our submission. We read carefully all reviewers comments and used this opportunity to submit response to Editor and reviewers. We have now addressed the issues raised by yourself and each reviewer and highlighted all changes throughout our manuscript for your convenience. The paper was re-checked for language errors by native English speaker and all technical mistakes are corrected throughout the article text and corrections are highlighted.

Reviewer 2 comments:
Reviewer 2: Authors of this Review for IJMS aimed to summarize aspects linking GLA players and a group of unrelated liver diseases. Authors should acknowledge that the story is much more complex, and for each of the conditions they have described there are several pieces of oceanic information. E.g., see how others tried to summarize the roles of the many players involved in GLA/microbiota and NAFLD:

1: Poeta M, et al . Gut-Liver Axis Derangement in Non-Alcoholic Fatty Liver Disease. Children (Basel). 2017 Aug 2;4(8).

2: Clemente MG, et al. Pediatric non-alcoholic fatty liver disease: Recent solutions, unresolved issues, and future research directions. World J Gastroenterol. 2016 Sep 28;22(36):8078-93

3. Nobili V,  et al. . Fighting Fatty Liver Diseases with Nutritional Interventions, Probiotics, Symbiotics, and Fecal Microbiota Transplantation (FMT). Adv Exp Med Biol. 2018 Dec 22.

Author: Dear reviewer, thank you very much for more than useful comments that helps us a lot with manuscript improvement. We respected them all and corrected the manuscript in accordance with your suggestions. We agree that liver diseases discussed in the review are not related, but they are diseases most commonly seen in every day practice of hepatologists and infectologists. In addition, these are diseases with whom we are the most familiar, as we are seeing patients with these diseases routinely every day, so we felt more competent to write review about these liver diseases than about other diseases. The idea of the review was to discuss relationship of GM and the most common liver diseases, not to focus on single diseases as it was done in the above-suggested manuscripts. Anyhow, these manuscripts helped us a lot to improve some segments of the paper.

Reviewer 2: On the other hand, the same treatment done for NAFLD should be reserved to their HBV, HCV, HCC, HE sections and so on. We understand that this might need a book. Therefore, in my opinion, Authors should focus better which is their MS's target. Once decided, they should be extremely precise and avoid putting crowds of information together.

Author: We agree that GM   role in different diseases including liver diseases is very complex and   requires many different data. We reorganized some parts of the manuscripts,   deleted and corrected some sentences and tried to avoid putting crowds of   information together and aimed to make it more readable.

Reviewer 2: The topic chosen is not new. Several innovative reviews capable of giving unifying hypotheses have been published and continue to be published on that issue. As submitted  in its present form the Authors' MS deals with a relevant topic, but then it appears  a mix of drops in the ocean, not inter/intra - linked each-other, which appear to have been selected without taking into account the endless explosion of very specific acquisitions in the field. The conclusion the readers can draw from this MS risks  therefore to be unhelpful to experts and to non-experts as well.

Author: The idea of the review was to focus on separate liver diseases and it’s relationship with gut microbiota as well as to point on potential therapeutical options that could include probiotics, prebiotics etc. There are a few similar reviews in the literature, but the advantage of the present review is that it is up-to date, includes discussion of novel published studies and explains pathogenesis of gut-liver axis and potential treatment options based on alterations in microbiota. In addition, this review is based on the individual liver diseases, which are the most common in clinical practise. The manuscript is mostly based on discussion on novel data, although many questions remain unanswered, but the manuscript opens space for original studies based on unanswered questions stated in conclusion.

Reviewer 2: Specific issues: A. AUTHORSHIP: In my personal opinion 12 Authors are too many for a review. Authors should give some stronger reason than that shown in the dedicated section of the MS.

Author: We agree that 12 authors is too much for the review, in general, but the present review is based on discussion of few different liver diseases and each and every author is responsible for one different part of the review, literature search and writing. We would like to explain that IJMS Guidelines for Authors did not propose allowed number of authors for review. Although it is not referent, we would like to kindly state that reviews with same number of authors have been already published in IJMS. We kindly ask you to agree with suggested number of authors.

Reviewer 2: B. TEXT READIBILITY: The text is difficult to read. 1.  Please indicate with headings/sub headings when you change topic. 2.  A few Summarizing Tables might be OK for improving the readibility.

Author: We agree that readability of the first version of text needed improvement. We modified the text as it is suggested, added headings and subheadings when the topic is changed. In addition, we prepared Tables that summarized the most important studies related to gut microbiota-associated liver diseases and therapeutic options for gut-microbiota alteration in liver diseases.

Reviewer 2: 3.Similarly, a Summarizing synoptic picture of GLA and liver diseases might help to follow better the issues which have been or are going to be treated in the text.

Author: Thank you very much for useful suggestion. We prepared summarizing synoptic picture of relationship of gut microbiota and liver diseases.

Reviewer 2: 4.PNALD, IBD, Cystic Fibrosis, PSC should be included among the liver diseases of interest:  e.g. see as others have done for celiac disease/GLA and liver : Marciano et al.  Dig Liver Dis. 201Feb;48(2):112-9.

Authors: Although these are important diseases and their connection with gut-microbiota is known, we did not chose to discuss these diseases as we are “expertized” in treating liver diseases, we are seeing such patients in every day clinical practice, and we do not feel relevant to discuss other diseases.

Reviewer 2: Line 108 to 115 is too detailed and appears out of the MS scope.  

Author: We agree that these sentences were too detailed and out of the scope of manuscript and gave impression that the focus is lost in this part. We corrected this segment and mostly deleted text from lines 108-115.

Reviewer 2: D.Ethanol: Authors please clearly specify that it is Endogenous EtOH.

Author: Dear reviewer, thank you for correction, we stated over the text that we thought on endogenous ethanol.

Reviewer 2: E. FXR: Bile acid synthesis, negative feedback inhibition through FXR, hepatic and ileal FXR  expression;  FXR  role in lipid and glucide metabolism;  dysbiosis elated abnormal bile acid modifications steps, change of FXR activation and signaling as causes of  cholestasis and leaky gut dependent liver changes should be better clarified  (see the enlightening MS of Leung DH, Yimlamai D. The intestinal microbiome and paediatric liver disease. Lancet Gastroenterol Hepatol. 2017 Jun;2(6):446-455.)

Author: Thank you very much for the useful suggestion. In accordance with that, we revised the section 3.3. The role of FXR in mediating the crosstalk between host and GM is now better clarified, taking into account the suggested reference (Please see the lines 226-253)

Round  2

Reviewer 2 Report

The MS appears much improved.

Now it is easier and more pleasant to read it.

I have some additional suggestions to improve further thenew parts  MS:

Figure/Scheme 1:

a. Please re-type the labels of the GI mucosa insert.

b. Also explain in the legend: NFKb, NPAA ,, etc.

Tables 1 and 2 :

a. organize  graphically the 2 tables more accurately 

b. select the references with better criteria (more recent RC T and/or meta-analysis)

c. quote the references (only) with their Number in the Reference section

d. (Logurecio = LoGuercio)

Abbreviation Lists:  should be in Alphabetical order

References: Be more consistent with Journals' names abbreviations. (e.g. refs ..., 62, 66, ..., )

Author Response

International Journal in Medical Sciences

12.01.2019.

Manuscript No. ijms-423157

“Gut-liver axis, gut microbiota and its modulation in the management of liver diseases: a review of the literature”

SECOND RESPONSE TO REVIEWER’S COMMENTS (2)

Dear Reviewer,

Thank you very much for the positive feedback and for more than useful comments. We would like to thank you for your valuable time, suggestions and comments with regards to our submission. We read carefully all comments and used this opportunity to submit response. We have now addressed the issues raised by you and highlighted in yellow all changes throughout our manuscript for your convenience. The paper was re-checked for language errors by native English speaker.

Reviewer’s comments and suggestions for authors

Reviewer: The MS appears much improved. Now it is easier and more pleasant to read it.

I have some additional suggestions to improve further the new parts MS:

Figure/Scheme 1:

a. Please re-type the labels of the GI mucosa insert.

b. Also explain in the legend: NFKb, NPAA ,, etc.

Author: Thank you very much once again for the support and for more than useful comments that helps us a lot with manuscript improvement. We corrected Figure and placed “GI mucosa” instead of “GM” and corrected abbreviation for NFKb. Appropriate abbreviations for GI, NFKb, LPS, NRL, TRL etc. are given below the figure. Abbreviations below the figure and tables are given in alphabetic order, as well as abbreviation list at the end of the manuscript text.

Reviewer: Tables 1 and 2:

a. organize  graphically the 2 tables more accurately 

b. select the references with better criteria (more recent RC T and/or meta-analysis)

c. quote the references (only) with their Number in the Reference section

d. (Logurecio = LoGuercio)

Author: Both tables are reorganized in more logical order, e.g. CHB, then CHC, then HCC etc. Also, we reorganized sentences in the row “Dysbiotic figures” with aim to make them uniform (e.g. always comes first explanation what is decreased, then what is increased etc.). References are quotes with their Numbers, in according to main reference list. The study done by LoGuercio et al. is not anymore cited as reference in the table, as it is not up-to-date reference, and it is replaced with more accurate one. In addition, one more reference dating from 2006 is replaced with more accurate one, dating from 2017. All new references are cited in the text as well (please see lines 367-369 and 436-441; reference no. 77,97,100).

Reviewer: Abbreviation Lists:  should be in Alphabetical order

Author: Abbreviations below the figure and tables are given in alphabetic order, as well as abbreviation list at the end of the manuscript text.

Reviewer: References: Be more consistent with Journals' names abbreviations. (e.g. refs ..., 62, 66, ..., )

Author: All technical mistakes are corrected in the references. We would like to use this opportunity to thank you very much for the indeed very useful suggestions and comments that significantly helped us to improve this review!

Our very best wishes,

Authors

Round  3

Reviewer 2 Report

The Authors followed this Reviewer's suggestions to improve further the quality and readibility of their MS, which now might appear of more interest for the Journal readers.

I have still 2 very minor changes to ask

A.  Reference 77. M, Y.Y.; ..... should be   77. Ma,Y.Y. ..........

B.  The headings "Tight Junctions" and "Disrupted tight junctions" INSIDE the figure 1 should be substituted with more readible new ones (copy and paste with Word)

Author Response

International Journal in Medical Sciences

13.01.2019.

Manuscript No. ijms-423157

“Gut-liver axis, gut microbiota and its modulation in the management of liver diseases: a review of the literature”

THE THIRD RESPONSE TO REVIEWER’S COMMENTS (2)

Dear Reviewer,

Thank you very much for the your positive comments that really improved our manuscript.

The Authors followed this Reviewer's suggestions to improve further the quality and readibility of their MS, which now might appear of more interest for the Journal readers.

I have still 2 very minor changes to ask

A.  Reference 77. M, Y.Y.; ..... should be   77. Ma,Y.Y. ..........

B.  The headings "Tight Junctions" and "Disrupted tight junctions" INSIDE the figure 1 should be substituted with more readible new ones (copy and paste with Word)

We made final corrections. Thank you for your patients and effort to support us.

Kind Regards,

Author